# The Kinetics of Pyrite Dissolution in Nitric Acid Solution

**DOI:** 10.3390/ma15124181

**Published:** 2022-06-13

**Authors:** Samaneh Teimouri, Johannes Herman Potgieter, Lizelle van Dyk, Caren Billing

**Affiliations:** 1School of Chemical and Metallurgical Engineering, University of the Witwatersrand, Private Bag X3, Wits 2050, South Africa; herman.potgieter@wits.ac.za (J.H.P.); lizelle.vandyk@wits.ac.za (L.v.D.); 2Department of Natural Sciences, Manchester Metropolitan University, Chester Street, Manchester M1 5GD, UK; 3School of Chemistry, University of the Witwatersrand, Private Bag X3, Wits 2050, South Africa; caren.billing@wits.ac.za

**Keywords:** pyrite, nitric acid, oxidative leaching, mechanism, mixed kinetic model, sulphur species

## Abstract

Refractory sulphidic ore with gold captured in pyrite has motivated researchers to find efficient means to break down pyrite to make gold accessible and, ultimately, improve gold extraction. Thus, the dissolution of pyrite was investigated to understand the mechanism and find the corresponding kinetics in a nitric acid solution. To carry this out, the temperature (25 to 85 °C), nitric acid concentration (1 to 4 M), the particle size of pyrite from 53 to 212 µm, and different stirring speeds were examined to observe their effect on pyrite dissolution. An increase in temperature and nitric acid concentration were influential parameters to obtaining a substantial improvement in pyrite dissolution (95% Fe extraction achieved). The new shrinking core equation (1/3ln (1 − X) + [(1 − X)^−1/3^ − 1)]) = kt) fit the measured rates of dissolution well. Thus, the mixed–controlled kinetics model describing the interfacial transfer and diffusion governed the reaction kinetics of pyrite. The activation energies (E_a_) were 145.2 kJ/mol at 25–45 °C and 44.3 kJ/mol at higher temperatures (55–85 °C). A semiempirical expression describing the reaction of pyrite dissolution under the conditions studied was proposed: 1/3ln(1 − X) + [(1 − X)^−1/3^ − 1)] = 88.3 [HNO_3_]^2.6^ r_0_^−1.3^ e^−44280/RT^ t. The solid residue was analysed using SEM, XRD, and Raman spectrometry, which all identified sulphur formation as the pyrite dissolved. Interestingly, two sulphur species, i.e., S_8_ and S_6_, formed during the dissolution process, which were detected using XRD Rietveld refinement.

## 1. Introduction

High-grade resources with easily extractable gold are becoming increasingly depleted. To fulfil the high demand for gold, owing to its versatile applications, extracting gold from low-grade and more challenging gold ores is desirable. Many of these sources are refractory in nature: they are not amenable to conventional cyanidation [1,2]. In general, three factors can contribute to making a gold ore refractory: (1) Very fine gold particles (<10 µm) enclosed in impermeable gangue minerals, often sulphides; such fine gold cannot be adequately liberated, even with fine grinding, and so remain inaccessible to the leaching solution. (2) Preg-robbing, which happens in carbonaceous ores by the adsorption of gold complexes onto a fine-grained carbonaceous mineral existing in the ore. (3) Copper–gold ores (complex ores) containing minerals, e.g., metallic copper, copper oxides, chalcocite, and bornite, can react with cyanide to form copper–cyanide complexes, which lead to a high cyanide consumption [3].

The degree of refractoriness depends on the severity of its causes in the gold ore, which can be one or a combination of these [4]. The most common reason for refractory ore is the invisible locked gold (chemically or physically) within the host minerals in the form of nanoparticles of native gold, colloidal particles, and solid solution [5]. The term “invisible gold” is used since it is difficult to observe with optical microscopy [6]. Pyrite and arsenopyrite are the predominant hosts for gold occluded in their structures [7].

As the gold is locked inside the sulphidic host mineral, it is necessary to break down the host matrix to expose the encapsulated gold. This justifies the pretreatment of the refractory gold ore before extracting gold to oxidise the impervious sulphidic mineral to obtain a permeable matrix for acceptable gold recovery [8]. Generally, the pretreatment is performed either pyrometallurgically or hydrometallurgically. The most common oxidative pretreatment methods are roasting [9], pressure oxidation [10], and bio-oxidation [11,12], and have also been investigated as options to destroy pyrite. Roasting requires high temperatures, e.g., 600 to 1000 °C, depending on the mineralogy of the ore, and is associated with releasing harmful gases, such as SO_2_, which pollute the environment if not captured. Pressure oxidation uses high pressure (4500 kPa) and temperature (≥250 °C), which pose safety risks, as well as high capital expenditure for specialised equipment and operating costs. Bio-oxidation typically demands a long processing time for pretreatment, namely, 4 to 8 days, depending on the ore [13,14,15,16].

Consequently, developing an oxidative pretreatment process with mild conditions, a low operating cost, and relatively fast kinetics is crucial. A hydrometallurgical process using nitric acid may be a potentially efficient way to oxidise and dissolve the sulphidic matrix of the host minerals, i.e., pyrite to expose the enclosed gold. While nitric acid is an oxidising acid that is capable of oxidising most metals, the emphasis of this study is on the oxidation of pyrite. Several investigations reported the oxidation of various minerals, such as chalcopyrite, galena, and arsenopyrite, using nitric acid [17,18].

In nitric acid (HNO_3_), the nitrogen (N) in NO_3_^−^ is in its highest oxidation state, which is +5; therefore, the NO_3_^−^ possesses strong oxidant abilities. This feature can be used for the oxidative pretreatment of refractory sulphidic ore, ultimately improving gold yield [19,20].

Nitric acid as an oxidising acid was the focus of some research investigating the kinetics of sulphidic gold ore dissolution. Gao et al. [13] investigated the kinetics of refractory gold concentrate ore (high sulphur and arsenic) with a diluted nitric acid solution. The ore dissolution was increased by increasing the nitric acid concentration and temperature and decreasing the particle size. The experimental results were evaluated using a shrinking particle model, and it was found that the reaction kinetics were controlled by the diffusion of NO_3_^−^ to the surface of the particles. The obtained activation energy was 12.3 kJ/mol in a 25% nitric acid solution. The kinetics of refractory sulphidic gold ore dissolution using nitric acid was studied by Rogozhnikov et al. [5] at temperatures of 70–85 °C. Increasing the temperature and nitric acid concentration improved the iron dissolution as an indication of sulphidic gold ore dissolution during 1 h of leaching. The new variant of the shrinking core model representing an interfacial transfer and diffusion fitted the results well. The reaction order concerning the nitric acid concentration was 0.84 and the obtained activation energy was 60.3 kJ/mol.

Although other researchers have investigated the dissolution of pyrite in other different mineral acids, such as H_2_SO_4_, HCl, and HClO_4_ [21,22,23,24,25], this investigation focused on HNO_3_. Nitric acid was chosen in preference to sulfuric acid and hydrochloric acid because it is the only acid that, during the dissolution process, yields a potentially recyclable product (nitric oxide and nitrous gas), which can be absorbed in a column and regenerated as a reagent (HNO_3_). This option clearly changes the economic prospects of any proposed process. Above all, HNO_3_ was investigated for the removal of pyrite contained in coal [26,27].

This research explored a wider range of experimental conditions than have been conducted to date. More than other research on this topic, it made use of a variety of solid analyses on the residue (SEM-EDS, XRD, XRD Rietveld refinement, and Raman spectroscopy) to examine changes in the morphology of the remaining pyrite grain and the produced product(s).

Therefore, this work attempted to describe the oxidative dissolution mechanism of pyrite in nitric acid by investigating the corresponding kinetics of the reactions. To attempt this, pyrite dissolution in nitric acid was studied at different leaching conditions by examining pyrite dissolution in the temperature range of 25 to 85 °C, concentrations of nitric acid from 1 to 4 M, various-sized fractions of pyrite from 53 to 212 µm, and different stirring rates. The selected temperature and acid concentration ranges employed covered a larger set of conditions than previous investigations, and could potentially also resolve the discrepancy in the vastly different activation energies reported in the earlier work referred to above. While the recovery of gold from refractory ores in which it is encapsulated in pyrite was the ultimate aim of this research, we endeavoured to first understand and focus on the dissolution of the pyrite only, rather than investigating any release of the gold. Once one understands how the pyrite can be broken down, it can be applied to refractory gold ores to establish how the gold recovery can be enhanced. Knowledge generated by this study could be used as a foundation to develop a nitric acid pretreatment process for sulphidic refractory gold ores.

## 2. Materials and Methods

### 2.1. Materials

The pyrite (FeS_2_) sample used in this study was obtained from Mintek, South Africa. Figure 1 displays the X-ray diffractogram (XRD) (Bruker, Karlsruhe, Germany) of the phase composition of the pyrite sample. Table 1 presents the chemical composition of the pyrite sample determined via X-ray fluorescence (XRF). To determine the particle size distribution (PSD), the pyrite sample was measured 3 times via laser diffraction (Mastersizer 2000). The average particle size and the standard deviation were determined and calculated to be d_10_ = 3.3 ± 0.6 µm, d_50_ = 18 ± 3.2 µm, and d_90_ = 77 ± 2.2 µm (Figure 2). The pyrite sample was dried in an oven and screened with stainless steel sieves to obtain different fraction sizes in the range of 53–212 µm using an electronic sieve shaker (Eriez model ES 200, Malvern, Worcestershire, UK).

### 2.2. Experimental Procedure

The oxidative leaching of pyrite with nitric acid was conducted in a 250 mL two-neck round-bottom flask as a reaction vessel, which was immersed in a water bath on a hot plate magnetic stirrer. A thermometer was positioned inside the reaction vessel to measure the temperature inside the leaching solution, and a water-cooled condenser was fitted to lead the nitrous gases to an absorption column. The experiment was temperature-controlled during stirring to provide a uniform pulp density. To examine the pyrite dissolution in nitric acid, a 10 g pyrite sample (particle size +75–106 µm) was added to the 200 mL of 3 M nitric acid (solid to liquid ratio of 1:20) when the temperature inside the leaching solution reached the desired temperature (25 to 85 °C). During the experimental runs, samples were withdrawn from the reaction vessel at regular time intervals (10, 20, 30, 60, 90, and 120 min) with a pipette and filtered using a 45 µm syringe filter. The iron content of the pregnant leached solution (PLS) was analysed using atomic absorption spectroscopy (AAS) (Agilent, Santa Clara, CA, USA) and used as an indication of pyrite dissolution. This formula was used to calculate the percentage of the extracted iron from the pyrite sample.
$${\%E}_{\mathrm{Fe}}=\frac{\mathrm{Amount}\;\mathrm{of}\;\mathrm{Fe}\;\mathrm{in}\;\mathrm{PLS}\;\mathrm{analysed}\;\mathrm{by}\;\mathrm{AAS}\;\left(\mathrm{mg}/L\right)\;\times \;\;\mathrm{Volume}\;\mathrm{of}\;\mathrm{solution}\;\left(L\right)}{\mathrm{Total}\;\mathrm{amount}\;\mathrm{of}\;\mathrm{Fe}\;\left(\mathrm{mg}/g\right)\;\times \;\;\mathrm{Amount}\;\mathrm{of}\;\mathrm{pyrite}\;\mathrm{used}\;\mathrm{in}\;\mathrm{the}\;\mathrm{experiment}\;\left(g\right)}\times \;100$$

After the experiments, the leached pulp was filtered using a Buchner funnel with filter paper. The residue (leached pulp) was washed with distilled water, dried in an oven at a temperature of 80 °C for 30 min, weighed, and analysed further for residue characterisation. The same procedure was followed for the other experiments using different nitric acid concentrations of 1–4 M, various size fractions of pyrite from 53 to 212 µm, and different stirring rates at a temperature of 65 °C to investigate their effect on pyrite dissolution in nitric acid solution.

### 2.3. Analysis

The pregnant solution taken at regular time intervals was analysed using atomic absorption spectroscopy (AAS) to obtain the amount of iron extracted. The compositions of solid residues were examined using scanning electron microscopy (SEM) (ZEISS Sigma, Manchester, UK), XRD, and Raman spectroscopy. SEM was performed using a ZEISS Sigma 300 VP equipped with a field emission gun and connected to “Smart SEM” software to examine the morphology of the leached residue. The sample preparation for SEM analysis was achieved by mounting a sample (pyrite/residue) on carbon tape fixed to an aluminium stub. The samples were coated with C and PdAu to facilitate imaging. The aluminium stubs were then inserted into the stage of the SEM and examined. The phase analysis was inspected with a Bruker D2 Phaser XRD spectrometer with a Co anode (wavelength = 1.79 Å) using a 2θ range of 20–90° with 0.0274° increment steps. Raman spectra were acquired using the 514.5 nm line of a Lexel Mode 95-SHG argon-ion laser and a Horiba LabRAM HR Raman spectrometer (Horiba, Lille, France) equipped with an Olympus BX40 microscope attachment. The incident beam was focused onto the sample using a 100 × LWD objective (N.A. = 0.80) and the backscattered light was dispersed via a 600 lines/mm grating onto a liquid nitrogen-cooled CCD detector. The data were acquired and analysed using LabSpec v5 software (version 5.64, Horiba, Lille, France). The power of the laser at the sample was 0.4 mW to ensure that there was no localised heating of the samples.

## 3. Results and Discussion

### 3.1. Mechanism of the Pyrite–Nitric Acid Dissolution

Pyrite dissolution in nitric acid is an electrochemical reaction that consists of anodic and cathodic reactions [20]. The anodic dissolution of pyrite and the cathodic reduction of the oxidant happen simultaneously on the pyrite particle’s surface. The driving force for these reactions to occur is the potential difference across the pyrite–solution interface [21,22]. The general reaction of pyrite with nitric acid (HNO_3_) can be expressed as
FeS_2(s)_ + 8 HNO_3(aq)_ → Fe^2+^_(aq)_ + SO_4_^2−^_(aq)_ + S^0^_(s)_ + 8 NO_2(g)_ + 4 H_2_O_(aq)_(1)

The anodic reaction (oxidation of pyrite) is given by
FeS_2_ + 4 H_2_O → Fe^2+^ + SO_4_^2−^ + S^0^ + 8 H^+^ + 8 e^−^(2)

The cathodic reaction (reduction in oxidant NO_3_^−^) is given by [18]
NO_3_^−^ + 2 H^+^ + e^−^ → NO_2_ + H_2_O(3)

In pyrite, when the bond between Fe^2+^ and S_2_^2−^ breaks, there is no change in the oxidation number of iron (Fe^2+^). This can be explained since iron is the electropositive element in FeS_2_, causing it to transfer to the solution as it is (without releasing an electron). Thus, the oxidation number of iron in the pyrite lattice was +2, and the released iron species (Fe^2+^_(aq)_) into the solution are still +2. It can, however, further oxidise to (Fe^3+^_(aq)_) in an oxidative medium. However, sulphur, which is the electronegative element, loses an electron(s) and undergoes recombining reactions [28,29]. Sulphur has an oxidation number of −1 in the lattice; hence, when the bond breaks, the shared electron from sulphur is released to form sulphur (S^0^_(s)_) and even sulphate (SO_4_^2−^_(aq)_) when seven to eight electrons are released by sulphur (Equation (2)). The released electrons are captured by the oxidant (NO_3_^−^), which is reduced to NO_x(g)_ (NO, NO_2_) after accepting the electrons (Equation (3)).

The proposed pyrite dissolution mechanism in nitric acid solution is illustrated in Figure 3, presenting the ions at the surface of the pyrite mineral; the species Fe^2+^_(aq)_, S^0^_(s)_, and SO_4_^2−^_(aq)_ being released into the bulk solution; and the oxidant (NO_3_^−^) taking the released electrons by S to form NO_x(g)_, i.e., NO_2(g)_ [21,28]. To keep the figure simple, the water molecules and the solvated ions were not included, just the main reducing and oxidising species were presented.

Figure 4 represents the energy band diagram of the pyrite–solution interface. Based on the applied potential (or potential of a solution), the Fermi level (E_F_) (which is the level of energy that lies between the conduction and the valance band) can have a slightly different level from the surface state. At a certain potential, the Fermi level can overlap the surface state, and this potential is known as the breakdown potential. This is the beginning of the anodic dissolution and leads to an anodic current [21,22]. After reaching the breakdown potential and even higher, electrons are transferred and removed from the surface bonds, leading to the breaking of the bonds, and ions moving across the Helmholtz layers to the bulk solution. At this point, the rate-limiting step in the dissolution kinetics is the transferring of electrons (charges) across the Helmholtz layers (space-charge layer). By further increasing the potential, the Fermi level does not overlay the surface state anymore. At this point, the speed of the electron transfers reaches its highest value, and the rate-limiting step is the movement of the produced ionic species across the Helmholtz layers to the bulk solution. Hence, this model is described as an interfacial kinetics model [21,22].

### 3.2. Effect of Temperature

The evaluation of the effect of temperature was conducted on pyrite dissolution in 3 M nitric acid, where the temperature varied from 25 to 85 °C. Temperature vastly influences the process because it affects the rate of the leaching process, i.e., both the diffusion and the reaction rate [13]. The results of this experiment are presented in Figure 5. There was less than 20% and 40% iron extraction at temperatures of 25 and 35 °C. However, a marked increase to over 70% of iron extraction occurred at 45 °C. This was followed by a moderate increase in iron extraction to 85% at 55 °C, and the most iron extraction of more than 95% at 75 °C and 85 °C.

### 3.3. Effect of Nitric Acid Concentration

The strong oxidative ability of nitric acid came from a concentration higher than 2 M in the solution. By definition, when the concentration of nitric acid increases, the [NO_3_^−^] concentration (oxidant) and the acidity [H^+^] of the solution increase, and therefore, ultimately, the oxidative power in the solution escalates [13]. This is explained from the standard electrode potential (E^0^) provided by NO_3_^−^ from nitric acid in Equation (4), and the corresponding Nernst Equation (5), as follows:NO_3_^−^ + 2 H^+^ + e^−^ → NO_2_ + H_2_O E^0^ = 0.80 V(4)
(5)$$E=E^{0}+\frac{0.059}{n}\;\log\frac{{\left[H^{+}\right]}^{2}\left[{\mathrm{NO}}_{3}^{–}\right]}{P_{{\mathrm{NO}}_{2}}{}_{.}}\;$$

A 1 M nitric acid solution, for example, contains [NO_3_^−^] = 1 M and [H^+^] = 1 M. By substituting E^0^ = 0.80 V, n = 1 (number of electrons), [NO_3_^−^] and [H^+^] 1 M, and the partial pressure of NO_2(g)_ 1 atm in the Nernst Equation (5), it would produce a half-reaction potential of 0.80 V, calculated as follows:$$E=E^{0}+0.059\log{\left[1\right]}^{2}\;\left[1\right]=0.80+2\times 0.059\times \;\log\;\left[1\right]=0.80\;V\;$$

One has to keep in mind that this calculation is an approximation because of activities (taking the activity coefficient together with the concentration into consideration) rather than only concentrations should be used in the equation. While the effect in a 1 M nitric acid solution will be small, it will increase as the solution becomes more concentrated to 4 M and deviates from an ideal solution. In a concentrated ionic solution, the activities can differ from concentrations by as much as 50% and are usually lower than the concentration values. Nevertheless, it is clear that increased values for the nitrate ion and hydrogen ion concentrations in the equation will still result in an increase in the calculated E value. It is unlikely that the vapour pressure in the denominator will change. Therefore, even taking into account a 50% lower value for the nitric and hydrogen ion concentrations in a 4 M concentration solution, it will still cause an increase in the calculated value of E to 0.853 V (higher potential). This explains the stronger oxidative power for pyrite dissolution in the more concentrated nitric acid solution, and thus the increase in the pyrite dissolution.

As illustrated in Figure 6, increasing the concentration of nitric acid escalated the iron extraction, with a marked increase occurring between 1 and 2 M from 45% to approximately 74% iron extraction. This was followed by another notable increase between 2.5 and 3 M from 75% to 85% iron extraction, but the increase was less notable between 3 and 4 M. The highest iron extraction of close to 95% occurred with 4 M nitric acid. Thus, it was found that a substantial pyrite dissolution occurred at concentrations above 2.5–3 M, where HNO_3_ had an adequately strong oxidising ability.

### 3.4. Effect of Particle Size

To determine the effect of the particle size of pyrite, leaching experiments were carried out with different particle size fractions from 53 to 212 µm in a 3 M nitric acid solution. As illustrated in Figure 7, iron extraction was enhanced by decreasing the particle size. The reason for the improvement when decreasing the particle size was that the chemical reaction happens on the surface of the pyrite particles. Therefore, the smaller the particle size, the bigger the available surface area for the reactions to take place [13]. The different particle sizes of pyrite contributed to a high Fe extraction of more than 75% to approximately 92%.

### 3.5. Effect of Stirring Speed

In heterogeneous solid–liquid reactions, particularly those controlled by diffusion, determining the influence of the stirring speed is important. Therefore, experiments were conducted with different stirring speeds to determine its effect on the pyrite dissolution. As illustrated in Figure 8, increasing the stirring speed increased the Fe extraction, and thus, pyrite dissolution. The stirring speeds (1000 to 4000 rpm) resulted in a satisfactory Fe extraction above 80% to close to 98%. Mass transfer resistance is present in a heterogeneous system, and hence, by increasing the stirring speed, the turbulence intensity surges, the thickness of the diffusion layer reduces, the mass transfer increases, and subsequently, the reaction kinetics improves [13].

### 3.6. Kinetic Model

Leaching is a heterogeneous reaction between solid particles and a liquid [30]. In the case of pyrite leaching in nitric acid solution, the reaction(s) took place at the interface of pyrite (solid phase) and nitric acid (liquid), and the leaching reaction could be expressed as follows:HNO_3(l)_ + FeS_2(s)_ → Products

Generally, during leaching, the following sequence of events occurs: the diffusion of the leaching agent (nitric acid) through a thin liquid film surrounding the pyrite particles; anodic and cathodic reactions occur on the surface of the pyrite particles (including charge transfer); then, the diffusion of the produced products, i.e., Fe^2+^, S^0^, SO_4_^2−^, and NO_x,_ into the bulk solution takes place. The slowest process(es) acts as a limiting step(s) and controls the kinetic model [30].

To determine the kinetics of pyrite dissolution in nitric acid, experimental results of the studied parameters were evaluated using the new equation proposed by Dickinson and Heal [31] in the shrinking core model. Further references indicated below provide additional confirmation of our choice. The Dickinson and Heal model defines the interfacial transfer and diffusion (mixed model) through the product as 1/3ln (1 − X) + [(1 − X)^−1/3^ − 1] = k.t. Here, X is the fractional conversion, t is time (min), and k is the rate constant (min^−1^). Thus, the mechanism of pyrite dissolution in nitric acid occurs, which consists of anodic and cathodic reactions occurring at the pyrite–solution surface (interfacial layer); then, the movement of the produced ionic species diffuses into the bulk solution and can be kinetically described by this equation.

Hence, the function of the equation 1/3ln (1 − X) + [(1 − X)^−1/3^ − 1] = k.t versus time was plotted to obtain the correlation coefficient (R^2^) for each temperature to establish its fitness. The resulting consistent straight line with R^2^ close to one for all studied temperatures (Figure 9) indicated that the interfacial transfer and diffusion through the product (mixed model) controlled the reaction kinetics of pyrite dissolution in the nitric acid medium.

The suitability of the other shrinking core model equations as a limiting step to control the kinetics of pyrite dissolution was also evaluated. They included diffusion through the product layer (1 − 3(1 − X)^2/3^ + 2(1 − X) = k.t), diffusion through the liquid film (X = k.t), and the surface chemical reaction (1 − (1 − X)^1/3^ = k.t). However, their R^2^ values were not close to one, making them unfit descriptions of the reaction. This could be attributed to the different stages in pyrite dissolution, which makes the interfacial transfer and diffusion mixed model a better fit.

This kinetic model agreed with the work by Rogozhnikov et al. [5], who investigated the kinetics of the dissolution of refractory sulphidic gold-containing concentrated ore in nitric acid solution in the temperature range of 70–85 °C. It was also in agreement with the more recent work by this group on the effect of pyrite and ferric ions on the kinetics of arsenopyrite leaching in nitric acid solution [1]. Moreover, the research by Li et al. [32] on the kinetics of cerium leaching from a mixed rare earth concentrate with nitric acid was also described by the new version of the shrinking core model based on the interfacial transfer and diffusion through the product.

The rate constant (k) was obtained from the slope of each straight line in Figure 9. Figure 10 represents the graph of *lnk* as a function of 1000/T, where the slope of this graph is equal to −E_a_/R. As shown in Figure 10, the slope changed over the studied temperature range, with a higher activation energy of 145.2 kJ/mol at the lower temperature (25–45 °C), and a lower activation energy of 44.3 kJ/mol at higher temperatures (55–85 °C).

The function of the equation 1/3ln (1 − X) + [(1 − X)^−1/3^ −1] = k.t versus time was also evaluated with the experimental results on pyrite dissolution in varied concentrations of nitric acid solution from 1 to 4 M. As was the case for varying temperatures, the new variant of the shrinking core model produced a straight line, which fit with R^2^ values close to one for the different nitric acid concentrations (Figure 11).

The slope of each straight line from Figure 11, at various HNO_3_ concentrations at a temperature of 65 °C, was plotted as *lnk* versus *ln*(*HNO*_3_) to determine the reaction order with regard to HNO_3_ (Figure 12). Based on the slope of the graph, the resulting empirical order with regard to HNO_3_ was 2.55.

The same process was performed with the obtained results on the pyrite dissolution with different particle size fractions from 53 to 212 µm. The plot of the equation vs. time produced straight lines. The slope of each straight line was plotted as *lnk* versus *ln*(*r*_0_) to determine an empirical order concerning r_0_, which was calculated to be −1.3, based on the slope (Figure 13).

By substituting the Arrhenius equation k = k_0_ e^−Ea/RT^ in the new shrinking core mixed-model equation of 1/3ln (1 − X) + [(1 − X)^−1/3^ −1)]) = k.t, we obtained its expression as 1/3ln (1 − X) + [(1 − X)^−1/3^ − 1)] = k_0_ e^−Ea/RT^.t. Based on the obtained values, the following semi-empirical expression for pyrite dissolution in HNO_3_ was derived for the high-temperature range (55–85 °C):1/3ln (1 − X) + [(1 − X)^−1/3^ − 1)] = k_0_ [HNO_3_]^2.6^ r_0_^−1.3^ e^−44280/RT^.t

As presented in Figure 14, all of the experimental results were used to evaluate the proposed semiempirical expression and also to determine the k_0_ from the slope (88.3) of the plotted graph. Therefore, the semiempirical expression for pyrite dissolution in HNO_3_ under the studied conditions could be expressed as
1/3ln(1 − X) + [(1 − X)^−1/3^ − 1)] = 88.3 [HNO_3_]^2.6^ r_0_^−1.3^ e^−44280/RT^.t

### 3.7. Characteristics of Solid Residue

The pyrite sample before leaching and after leaching (residue) with nitric acid was examined with SEM-EDS. The SEM-EDS of pyrite before leaching detected 33.5% and 29.2% (weight%) of S and Fe, respectively. However, on the residue, the amount of sulphur increased to 49.5% and the Fe decreased to 4.5% owing to the formation of sulphur as the result of the oxidative dissolution of pyrite (Figure 15). Figure 16 shows the SEM images of the residue at 25, 45, 65, and 85 °C after leaching in 3 M nitric acid for 120 min. As the temperature increased, more pyrite particles dissolved, leaving fewer pyrite particles at the higher temperatures of 65 and 85 °C. This confirmed the positive effect of increasing the temperature on the dissolution of pyrite. Moreover, due to the oxidative dissolution of pyrite in nitric acid, sulphur formed as an amorphous product around the pitted pyrite grain. Therefore, in the nitric acid medium, the surface of the pyrite grain became rough and pitted with cavities (Figure 17) and the oxidative dissolution led to the formation of sulphur, as seen in the images.

The pyrite samples were examined with XRD before and after leaching in 3 M nitric acid at different temperatures of 25, 45, 65, and 85 °C. As illustrated in Figure 18, by increasing the temperature, the intensity of the pyrite peaks decreased owing to the pyrite dissolution. Interestingly, a new sulphur phase appeared, confirming the sulphur formation seen in the SEM images.

Moreover, the XRD Rietveld refinement inspected the solid residue qualitatively and quantitatively by detecting each phase and its amount at different temperatures. Table 2 summarizes the results for the XRD Rietveld refinement for different temperatures. Hence, after leaching in 3 M nitric acid at 65 °C, 11.3% (mass percent) and 0.09 (mole fraction) of pyrite remained while producing mostly sulphur (S_8_) and sulphur epsilon (S_6_) with 74% and 10.7%, respectively. However, at 85 °C, the pyrite was mostly dissolved (only 5% (mass percent) and 0.04 (mole fraction) remained) and 84% sulphur (S_8_) and 11% sulphur epsilon (S_6_) were produced as products of the oxidative dissolution. The mole fractions of pyrite indicated in brackets in Table 2 further confirmed how incongruent the dissolution was.

Figure 19 illustrates the spectra for the Raman analysis of the solid residue of pyrite after leaching in 3 M nitric acid at various temperatures. The Raman laser contacted each residue at four spots (a, b, c, and d) and detected both pyrite and sulphur at all studied temperatures. To see the pyrite and sulphur shifts clearly, their Raman shifts were depicted separately as the last two graphs presented in Figure 19. As depicted in the spectra, pyrite had three shifts at frequencies of approximately 343, 380, and 430 cm^−1^, and sulphur had three strong and two weak shifts situated at 154, 218, 247, 439, and 473 cm^−1^, respectively. The positions of the pyrite and sulphur shifts in the Raman analyses agreed with research by Xia et al. [33] and Toniazzo et al. [34]. Xia et al.’s [33] work focused on the surface analysis of sulphur species on oxidised pyrite, while Toniazzo et al.’s [34] work focused on sulphur speciation and quantification at the surface of pyrite. The Raman analysis also confirmed the formation of the sulphur product as the result of oxidative leaching of pyrite in nitric acid.

## 4. Conclusions

The oxidative dissolution of pyrite in nitric acid solution was investigated by altering influential leaching parameters. This study found the following:▪Pyrite dissolution at temperatures higher than 45 °C and up to 85 °C was between 72% and 95% after 120 min leaching in 3 M HNO_3_.▪At nitric acid concentrations between 2.5 M and 4 M, HNO_3_ had a powerful oxidising ability, resulting in an ample pyrite dissolution of between 74% and 95%.▪Reducing the fraction size of pyrite from +150–212 µm to +53–75 µm led to an increase in iron extraction from 75% to approximately 92%.▪Increasing the stirring speed from 1000 rpm to 4000 rpm resulted in Fe extraction of 80% to 98%.

The proposed new version of the shrinking core model, i.e., (1/3ln (1 − X) + [(1 − X)^−1/3^ − 1)]) = k.t), fit the kinetic results well. Thus, the mixed−controlled kinetic model describing the interfacial transfer and diffusion governed the reaction kinetics of pyrite dissolution in a nitric acid solution.

The activation energies (E_a_) of pyrite dissolution at lower temperatures (25–45 °C) and higher temperatures (55–85 °C) were calculated as 145.2 kJ/mol and 44.3 kJ/mol, respectively.

The semiempirical expression for pyrite dissolution in HNO_3_ under the studied conditions could be formulated as
1/3ln (1 − X) + [(1 − X)^−1/3^ − 1)] = 88.3 [HNO_3_]^2.6^ r_0_^−1.3^ e^−44280/RT^.t

▪The SEM images of the residue at different temperatures showed the surface of the pyrite grain became rough and pitted with cavities, and the oxidative dissolution in the nitric acid led to the formation of sulphur as the product.▪The XRD pattern of pyrite before leaching and the residue of leached pyrite in 3 M nitric acid at varied temperatures revealed that the intensity of the pyrite peaks decreased with an increasing temperature due to the pyrite dissolution and a new sulphur phase appeared.▪The XRD Rietveld refinement identified two sulphur species, i.e., S_8_ and S_6_, which formed during the dissolution process.▪The Raman analysis also confirmed the formation of sulphur at all the studied temperatures.

## Figures and Tables

**Figure 1 materials-15-04181-f001:**
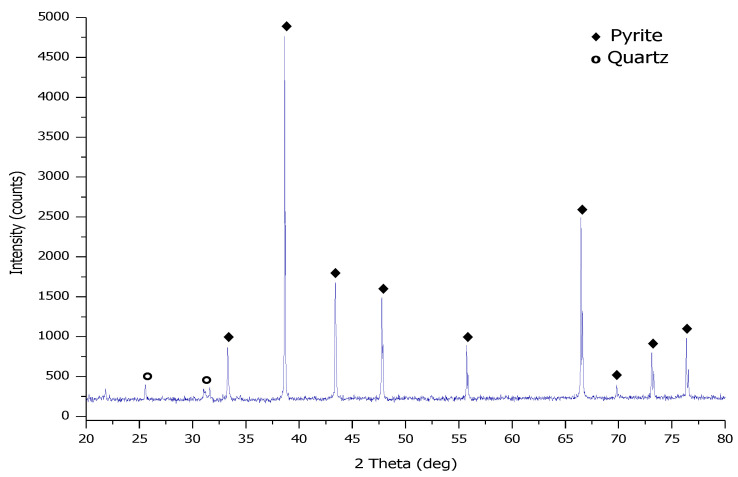
The XRD pattern of the phase composition of a pyrite sample.

**Figure 2 materials-15-04181-f002:**
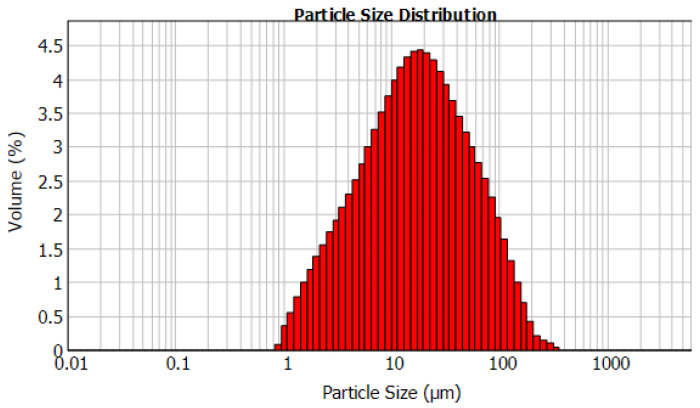
The particle size distribution (PSD) of the pyrite sample.

**Figure 3 materials-15-04181-f003:**
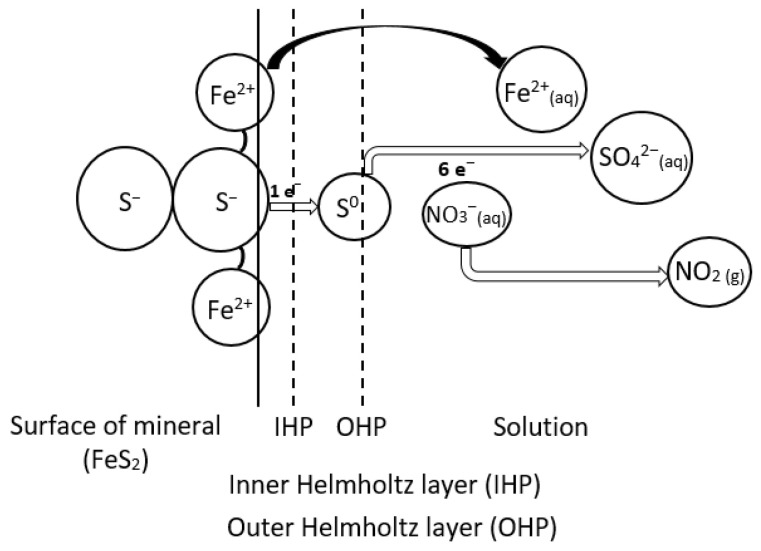
The proposed pyrite dissolution mechanism in nitric acid solution.

**Figure 4 materials-15-04181-f004:**
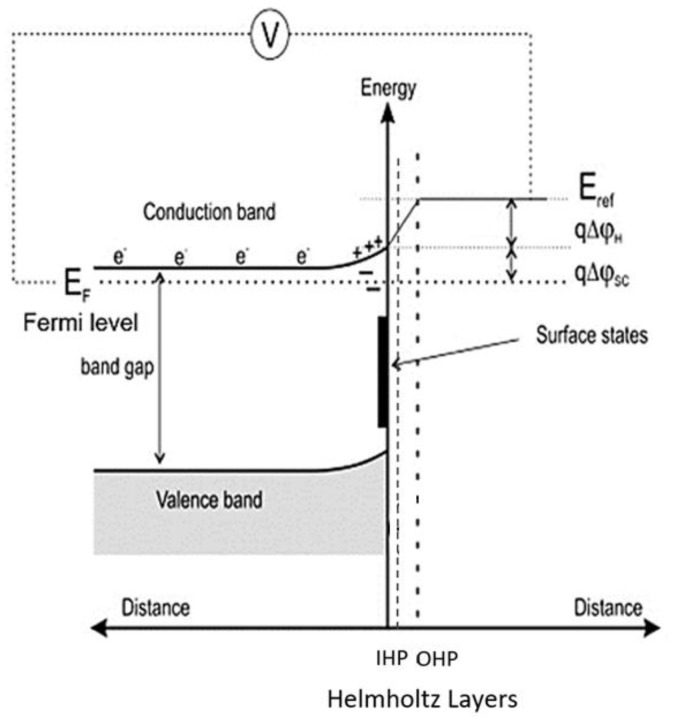
The energy band diagram of a pyrite–solution interface adopted from [21].

**Figure 5 materials-15-04181-f005:**
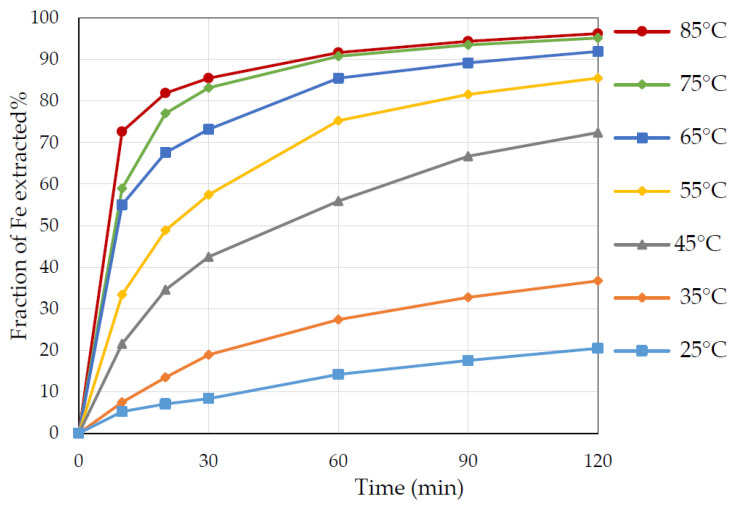
The extraction of iron from pyrite at different temperatures from 25 to 85 °C. Conditions: 3 M nitric acid, S/L ratio 1 to 20, particle size +75–106 µm.

**Figure 6 materials-15-04181-f006:**
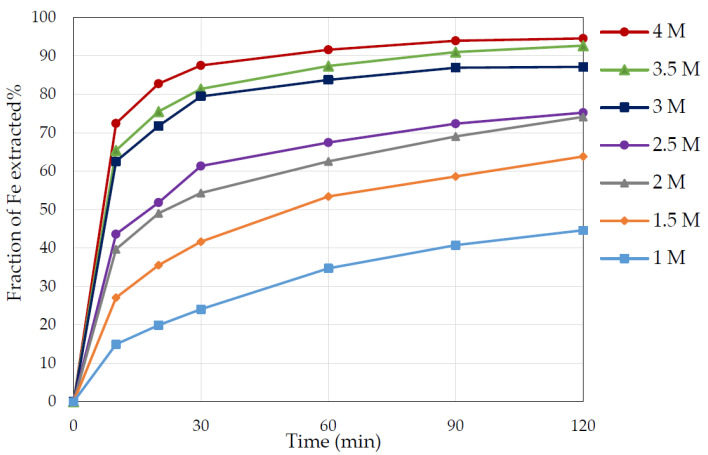
Time dependence of iron extraction from pyrite with different nitric acid concentrations (1–4 M). Conditions: temperature 65 °C, S/L ratio 1 to 20, particle size +75–106 µm.

**Figure 7 materials-15-04181-f007:**
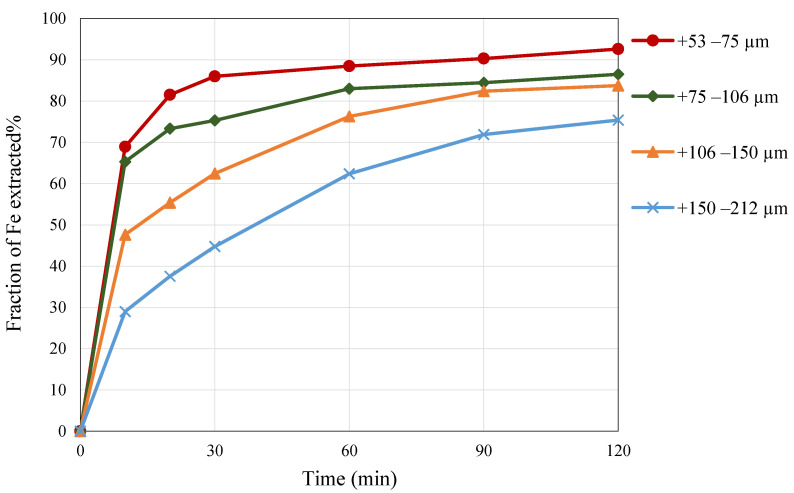
The effect of particle size on Fe extraction as a function of time. Conditions: 3 M nitric acid, temperature 65 °C, S/L ratio 1 to 20.

**Figure 8 materials-15-04181-f008:**
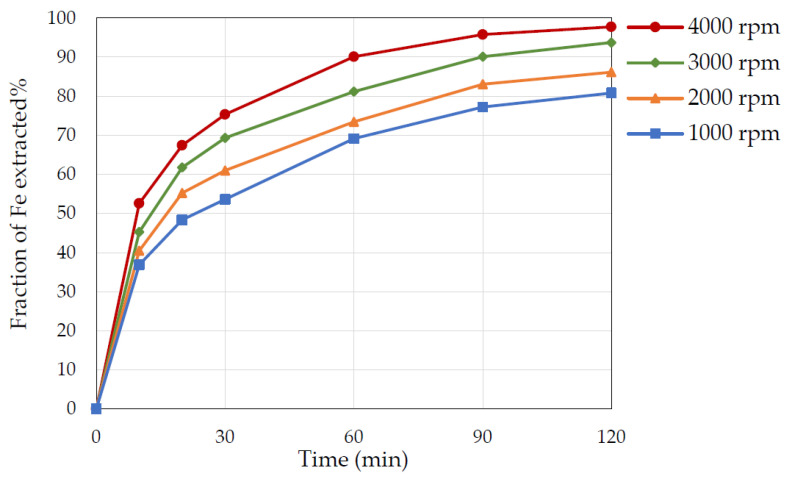
The effect of stirring speed on Fe extraction as a function of time. Conditions: 3 M nitric acid, temperature 65 °C, S/L ratio 1 to 20, particle size +53−75 µm.

**Figure 9 materials-15-04181-f009:**
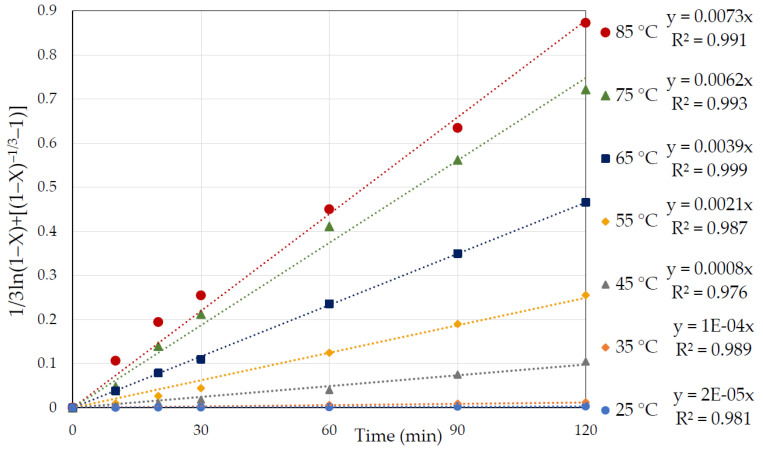
Graphs of 1/3ln (1 − X) + [(1 − X)^−1/3^ −1] = k.t vs. time (min) at various temperatures of 25 to 85 °C for the dissolution of pyrite in 3 M nitric acid.

**Figure 10 materials-15-04181-f010:**
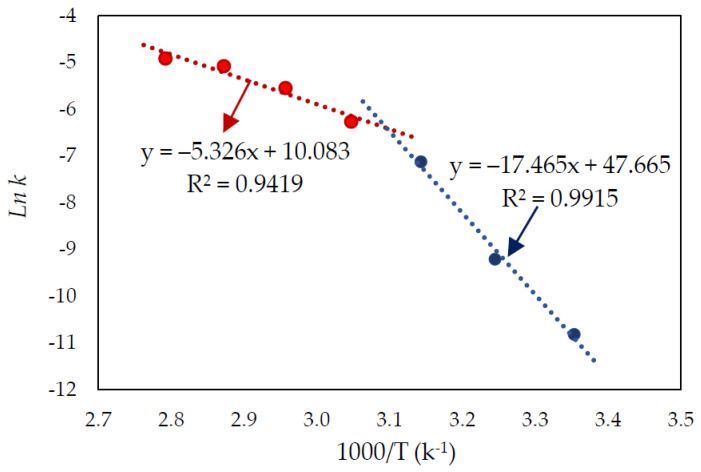
Arrhenius plot of lnk vs. 1000/T (k^−1^).

**Figure 11 materials-15-04181-f011:**
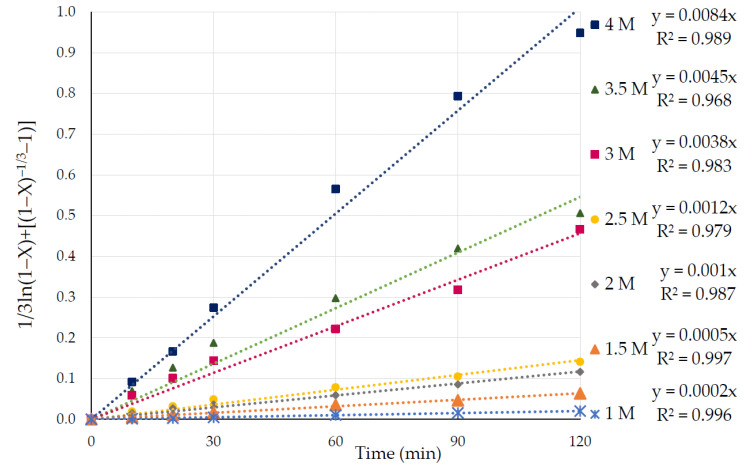
Graph with curves of 1/3ln (1 − X) + [(1 − X)^−1/3^ − 1)] vs. time (min) at different concentrations of HNO_3_ at a temperature of 65 °C.

**Figure 12 materials-15-04181-f012:**
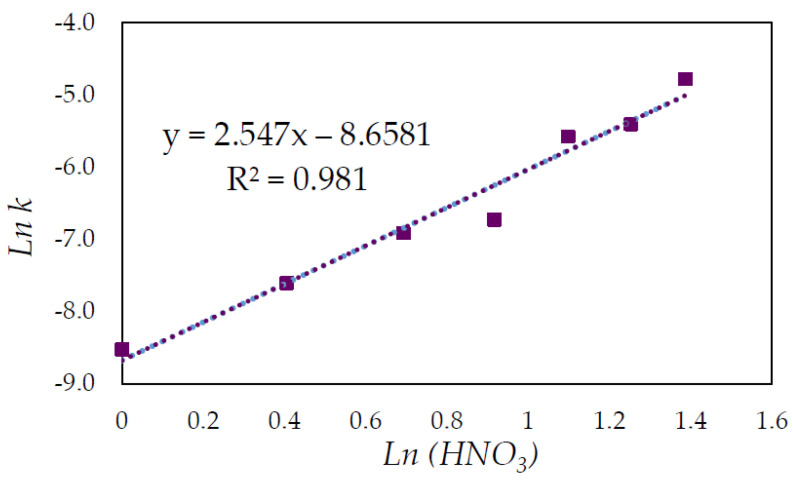
Plot of lnk as a function of *ln*(*HNO*_3_).

**Figure 13 materials-15-04181-f013:**
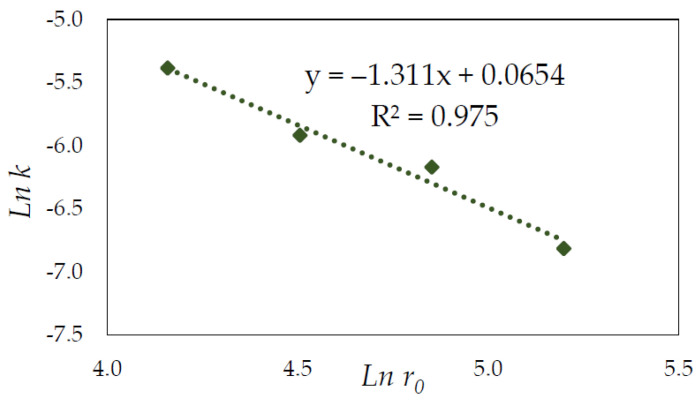
Plot of lnk as a function of *ln*(*r*_0_).

**Figure 14 materials-15-04181-f014:**
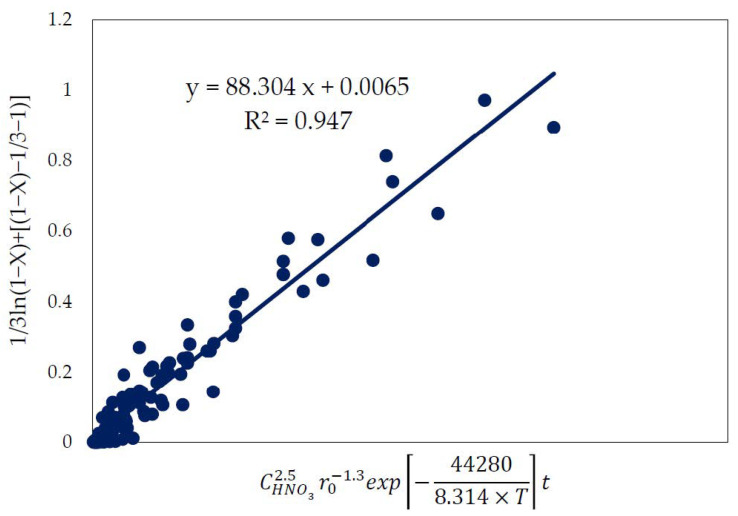
Relation of the applied kinetic equation and the acquired semiempirical expression.

**Figure 15 materials-15-04181-f015:**
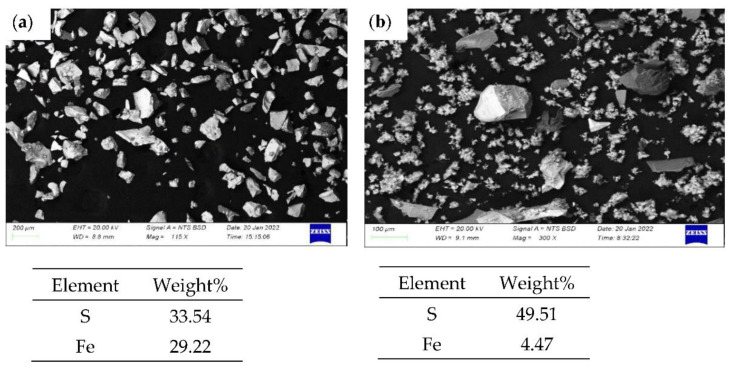
The backscattered SEM-EDS images of (**a**) pyrite before leaching and (**b**) after leaching in nitric acid.

**Figure 16 materials-15-04181-f016:**
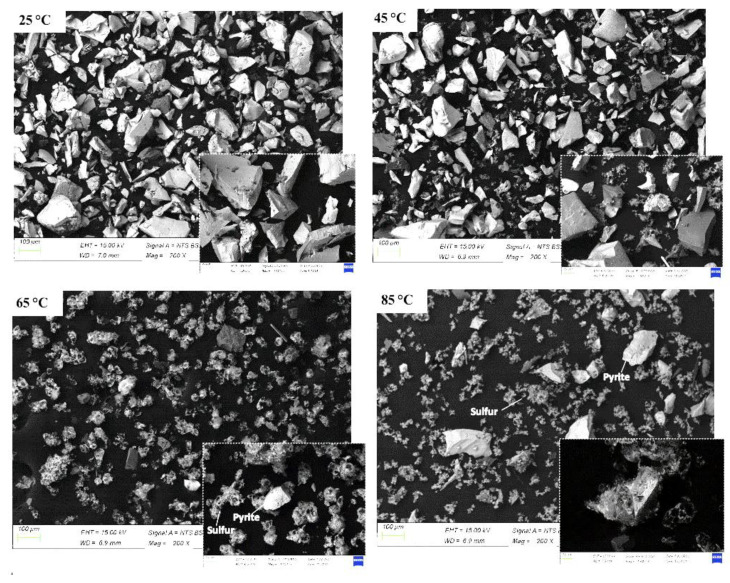
SEM images of pyrite residue at different temperatures of 25, 45, 65, and 85 °C.

**Figure 17 materials-15-04181-f017:**
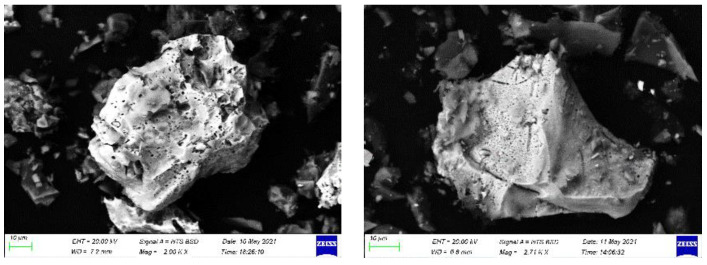
SEM images of leached pitted pyrite particles in 3 M nitric acid at a temperature of 65 °C.

**Figure 18 materials-15-04181-f018:**
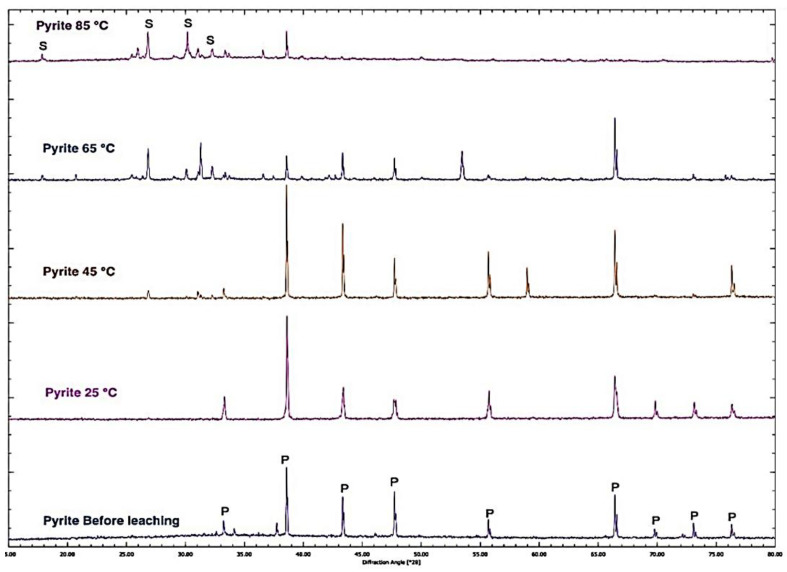
The XRD pattern of pyrite samples before leaching and the residue of leached pyrite in 3 M nitric acid at different temperatures of 25, 45, 65, and 85 °C.

**Figure 19 materials-15-04181-f019:**
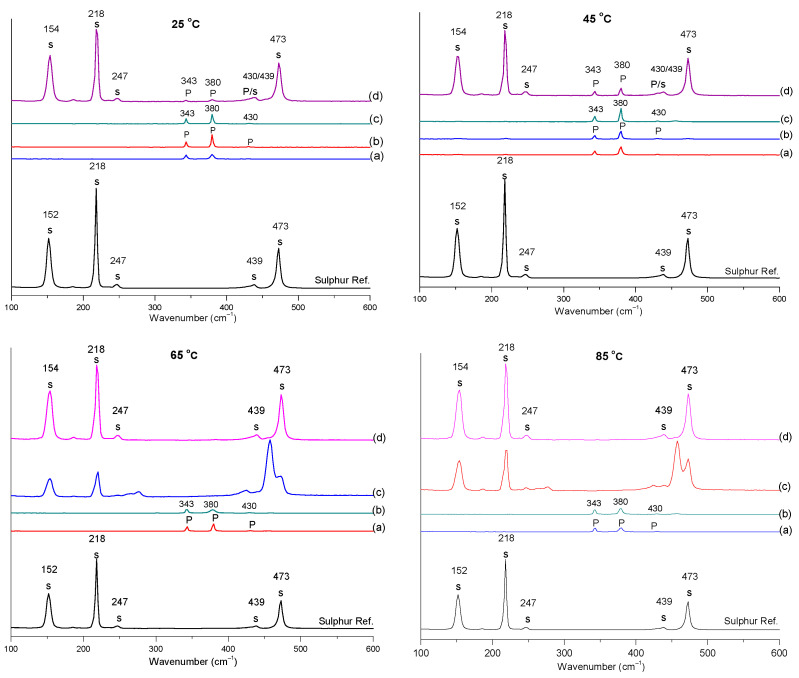
The Raman spectra of pyrite residue leached in 3 M nitric acid at various temperatures of 25, 45, 65, and 85 °C, showing the Raman spectra of pyrite and sulphur separately.

**Table 1 materials-15-04181-t001:** The chemical composition of the pyrite sample as determined using XRF.

Chemical Composition	Al_2_O_3_	SiO_2_	TiO_2_	CaO	S *	FeS_2_	LOI
Wt%	2.25	4.19	0.45	1.38	13.41	69.12	9.16

* Measured using a Leco instrument (not bound in the pyrite).

**Table 2 materials-15-04181-t002:** The XRD Rietveld refinement for different temperatures.

Temp./Phases	Pyrite (FeS_2_) (Mole Fraction)	Quartz (SiO_2_)	Sulphur (S_8_)	Sulphur Epsilon (S_6_)
Leaching at 25 °C	96.5% (0.8)	0.5%	3.1%	–
Leaching at 45 °C	85.1% (0.7)	3.8%	11.1	–
Leaching at 65 °C	11.3% (0.09)	3.3%	74.7%	10.7%
Leaching at 85 °C	5% (0.04)	–	84.0%	11.0%

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
