# Peer review of "The Kinetics of Pyrite Dissolution in Nitric Acid Solution"

_materials, 2022, doi:10.3390/ma15124181_

Round 1

Reviewer 1 Report

The author studied the dissolution kinetics of pyrite in a nitric acid solution to find efficient means in improving gold extraction efficiency by breaking down pyrite structure. This research is valuable, but some revisions must be required before publication.

Point #1: Line 94, the reaction kinetics only showed the oxidative dissolution behavior of pyrite, but its mechanism was studied in combination with other means, such as XRD and Raman.

Point #2: XRF data of pyrite should be provided.

Point #3: The equation for calculating the fraction of Fe extracted should be given.

Point #4: In section 3.1, Figure 3 and the corresponding content are not more meaningful. The authors can of course provide corresponding references to readers who are interested in the corresponding mechanism, but it is of little significance in this paper. Furthermore, OHP is the layer in which hydrated ions are located, charge transfer generally does not undergo in this layer but in the inner Helmholtz layer (IHP). Therefore, the authors should distinguish the inner/outer Helmholtz layer.

Point #5: Particle size has a significant effect on the reaction kinetics and dissolution efficiency of pyrite. Therefore, the authors should report the particle size range of pyrites used in the chart header of all experiments. The same goes for other conditions.

Point #6: In Figure 4 and 5, as well as Figure 6 and 7, some data of iron extraction efficiency can not match the data of mass of residue and dissolved pyrite. There are some problems and the conclusions are unreliable. In addition, Figure 5 and 7 are redundant.

Point #7: Line 238-240, difficult to understand.

Point #8: Line 300, check the sentence.

Point #9: In Page 11, from the value of the activation energy and the variation of the activation energy with temperature, it can be concluded that the dissolution reaction of pyrite is controlled by the chemical reaction. Is it right?

Point #9: Provide the preparation conditions of the sample shown in Figure 16.

Point #10: In Figure 17, the SEM image of pyrite obtained at 45oC (dissolution efficiency exceeds 70%) is significantly different from 65 oC but similar to that of 25 oC (dissolution efficiency is about 20%). SEM images and scales are not clear. It is difficult to identify particles.

Point #11: In Figure 19, the XRD patterns of pyrite obtained at 45 oC and 65 oC is quite different from raw ore and treated at 25 oC. The author preferably provides the standard diagram and card number of S species and pyrite. The legend located in the upper right corner of the Figure should be removed.

Point #12: In Table 1, the XRD Rietveld refinement of the sample treated at 85 oC conveys the wrong information, while in fact, about 5% of the pyrite remains, as shown in Figure 4. Sulphur content is 11.1%.

Point #13: Line 416 to 419, this sentence is complicated. Rephrase.

Point #14: In Figure 20, a, b, c, and d should be assigned to? There is a small error in the unit of wavelength.

Point #15: Merging Figure 21 into Figure 20.

Point #16: The Conclusions are verbose and should be streamlined.

Author Response

Dear Reviewer,

I appreciate your constructive comments/suggestions. They made this manuscript a better version.

Best regards,

Samaneh Teimouri

Reviewer 2 Report

This is a very well-written manuscript and it was a pleasure to read.  The topic is interesting and eventually could have a high impact for gold extraction from pyrite ores.

I really only have a few major criticisms:

1. Section 3.1 seems out of place.  It is well-written and informative, but it has nothing to do with the rest of the results or their interpretation.  The remainder of the results section uses pretty standard kinetic analysis that would be just as applicable for ionic crystal dissolution, so why should there be so much detail about the electronic structure of pyrite?   I see two possibilities for improvement:  (a) eliminate most or all of Section 3.1, or (b) modify the paper to take better advantage of your knowledge of the oxidation-reduction reactions.  For example, why not apply an electric potential and characterize how that affects the kinetics, then interpret the results in light of the redox processes?

2. The solutions have pretty high ionic strength and so the use of concentrations in Eq. (5) and elsewhere is not appropriate.  You need an ionic activity model, and for ionic strengths as high as you are using, probably only the Pitzer ion interaction model will provide acceptable accuracy.

3.  You noted a major change in apparent activation energy in Fig. 11, which is a major observation and has implications for a change in the rate-controlling mechanism.  But you make almost no analysis of this except to report the two different activation energies.

4. Shrinking core models are numerous and common, varying in their details and assumptions.  Your kinetic analysis relies almost exclusively on a particular variant, so you really should give more details than a citation to a paper by Dickinson and Heal.

5.  In Section 3.7 you analyze the solid residue but you use mass percent of iron and sulfur.  Why not use mole fractions?  That would give a more straightforward indication of how incongruent the dissolution process is.

Author Response

Dear Reviewer,

Thanks for your encouraging feedback and constructive comments/suggestions.

Best regards,

Samaneh Teimouri

Reviewer 3 Report

See attachment documet

Author Response

Dear Reviewer,

I appreciate the valuable points that you bring out in this manuscript to improve it.

Best regards,

Samaneh Teimouri

Round 2

Reviewer 1 Report

Comments have been attached to the file.

Author Response

Dear Reviewer,
I truly appreciate your time and attention for improvement of this paper.

Comments and response:
Point #1: For XRF data. The Fe exists as Fe(II) in pyrite, so the content of Fe and S is suggested to be given as FeS2 instead of trivalent iron oxides and S.
- The amount of pyrite (FeS2) was calculated based on the values from XRF and been added to the XRF Table (1).
Point #2: The calculation equation of Fe extracted fraction should be given in the article.
- The Formula for the calculation of the percent of extracted Fe was added to the section 2.2. (Experimental Procedure).
Point #3: I'm sorry for misleading the author, but the wavenumber unit of Raman spectra is cm-1 not Cm-1.
- The wavenumber unit of Raman spectra was corrected to cm-1.
Point #4: I still support that Figure 20 is redundant as the author has clearly marked these peaks of sulphur and pyrite.
- I added the Raman shifts of pyrite and sulfur to Figure 19, for clarity.

Reviewer 2 Report

The authors addressed some of my comments in a perfunctory way.  However, they still have not addressed the problem with using concentrations as proxies for the activities in a nonideal solution.  I understand if they cannot or do not want to calculate activities for this paper.  However, the authors must at least acknowledge the problem and estimate the errors in E that will arise in Eq. (5) as a result of neglecting this effect.  In a concentrated ionic solution, the activities will generally differ from the concentrations by as much as 50 % (usually being lower than the concentrations).  So they could at least make an assumption of the activities of [H+] and [NO3-] each being 50 % lower than the concentrations (the vapor pressure in the denominator will not change since the vapor probably is ideal) and then propagate the uncertainties using standard error analysis to estimate their uncertainties in the values of E.

Author Response

Response to the reviewer’s comment round 2
We thank the reviewer for his comment regarding the effect of the ionic strength of the concentrated solution on the calculated potential in the Nernst equation. However, as the purpose was merely to illustrate that an increase in the nitric acid concentration results in an increase of the oxidative power of the solution, we did not consider it necessary to apply the suggested ionic strength model recommended for an accurate calculation of the increase. Nevertheless, we have, as recommended by the reviewer, added additional comments to the paper in the relevant paragraph (highlighted in yellow) and illustrated the effect of a 50% error in the values of the concentrations of the ions used to calculate the potential value to show that an increase occurred in the oxidative potential of the more concentrated solution which would explain the increase in the pyrite dissolution.
